# Quantitative research on the efficiency of ancient information transmission system: A case study of Wenzhou in the Ming Dynasty

**Yukun Zhang**[1☯], **Bei Wu**[1☯]*, **Lifeng Tan**[1☯]*, **Jiayi Liu**[2]

1 Key Laboratory of Information Technology for Architectural Heritage Inheritance of the Ministry of Culture and Tourism of China, School of Architecture, Tianjin University, Tianjin, China, 2 School of Marine Science and Technology, Tianjin University, Tianjin, China

☯ These authors contributed equally to this work.
* wubei@tju.edu.cn (BW); tanlf_arch@163.com (LT)

**Data Availability Statement:** All relevant data are within the paper and its Supporting Information files.

**Funding:** This work is supported by the National Natural Science Foundation of China (Grant No.

## Abstract

In ancient China, an unobstructed, convenient and efficient transmission system nationwide was established for long-distance transmission of information. The transmission system works to different degrees in different regions, which is an important index to measure the interregional information level. Yet, some minor differences, may not be easily sensed by people subjectively. Identifying and quantifying the influences of information transmission efficiency is the best way to solve this problem. Based on the historical information map visualized by ArcGIS software, this study established a hierarchy evaluation model suitable for the analysis of ancient information transmission efficiency from three aspects of Wei-Suo system, beacon system and post system. The information transmission systems in five different regions of Wenzhou in the Ming Dynasty were quantitatively explored respectively. The results break through the qualitative conclusions of the general studies, and find out that the overall information transmission efficiency of Wenzhou in Ming Dynasty was strong in coastal, northern and southern regions, but weak in inland and central regions, which was closely related to the geographical environment and military defense demands in coastal areas of the Ming Dynasty. The model is proven to greatly contribute to judging the spatial configuration of ancient information transmission system in different regions, and provides a new idea for the study on ancient information transmission system.

## Introduction

Spatial and temporal distances are always noticeable when it comes to information transmission. To cross the distance, delivery tools are indispensable [1]. In ancient China, the government set up the transmission system to ensure the dissemination of information, such as building beacon towers and setting off smoke, driving horses and cattle, running on foot, renovating roads, constructing post stations, etc. People relied on these ways to achieve the purpose of disseminating decrees and reporting military movements, which contributed greatly to the consolidation of feudal regime [2].

52078324 and 51778400), Major Research on Philosophy and Social Sciences of the Ministry of Education of China (Grant No. 19JZD056 and 18JZD059). The funders had no role in study design, data collection and analysis, decision to publish, or preparation of the manuscript.

**Competing interests:** The authors have declared that no competing interests exist.

The earliest organized communication behavior in China originated in the Shang Dynasty. At that time, there were special couriers who delivered information and were called Zhi in oracle bone inscriptions. By the Zhou Dynasty, a relatively regular transmission system was formed, and beacon towers had become a formal system. After Zhou, the transmission system was growing mature day by day, which was called "the vein of the nation" [3]. Thus the number of post stations increased greatly in Sui, Tang and Song Dynasties, and relevant laws and regulations were issued. In the Yuan Dynasty, post stations were developed on a large scale because of the war, and a transmission network centered on Dadu (大都) was established to lead to the whole country [4].

The transmission system of Ming Dynasty inherited from that of the Yuan Dynasty. On the 22nd day after the establishment of the Ming Dynasty, Emperor Zhu Yuanzhang (朱元璋) ordered the rectification and restoration of post stations throughout the country [3], and built beacon towers along the border to transmit military information. Compared with the transmission system of the previous generations, the laws on the transmission system in the Ming Dynasty were more completed, which had made specific provisions on the organization, network distribution, mileage time limit and funding quota. In addition, the post roads in the border areas were hewed out, which made the coverage of the transmission system larger. Therefore, after successive dynasties of evolution, the information transmission system has been very developed in the Ming Dynasty. By the middle of the Ming Dynasty, an excellent transmission network had been established nationwide, which penetrated into the politics, economy, military, nationality, culture, laws and other fields. Most of the early studies were conducted from different perspectives of history, such as organization form [5, 6], management administration [7–9], functions [10, 11] and development [12–14]. Recently, some scholars began to analyze the transmission facilities of a certain kind or in some areas geographically [15–18]. However, due to it's complex constitution, large quantities, and rough descriptions in historical records, overall quantitative analysis of Ming's information transmission system has always been a thorny issue.

In order to practice quantitative research method under the influence of multiple factors, Wenzhou (温州), a coastal city of Ming Dynasty, was selected as the study area. The information transmission efficiency of Wenzhou was tentatively investigated, presenting people with ancient information transmission system hidden in local chronicles, historical records and maps in a more clearly manner. In addition to evaluating the efficiency of information transmission, this study also attempts to find out whether the distribution of the transmission system was directly related to coastal defense demands, thus measuring the military information level of the Ming Dynasty.

## Materials and methods

### Study area

Wenzhou is located in the southeast of Zhejiang (浙江) Province, bordering the East China Sea in the east and Fujian (福建) Province in the south. In the Ming Dynasty, there were five counties in Wenzhou, namely Yongjia (永嘉) County, Yueqing () County, Rui'an () County, Pingyang (平阳) County and Taishun (泰顺) County, as well as a prefectural city situated in Yongjia County, which was the then highest administrative unit of Wenzhou (Fig 1). From the military point of view, Wenzhou had always been a gateway to Zhejiang, and the only route from Zhejiang to Fujian. Once Wenzhou fell, the military situation in the whole southeastern coastal area would be seriously affected. Moreover, Wenzhou was one of the main landing sites for Japanese pirates to invade Zhejiang Province. Therefore, the Ming government attached great importance to the construction of Wenzhou's information transmission system.

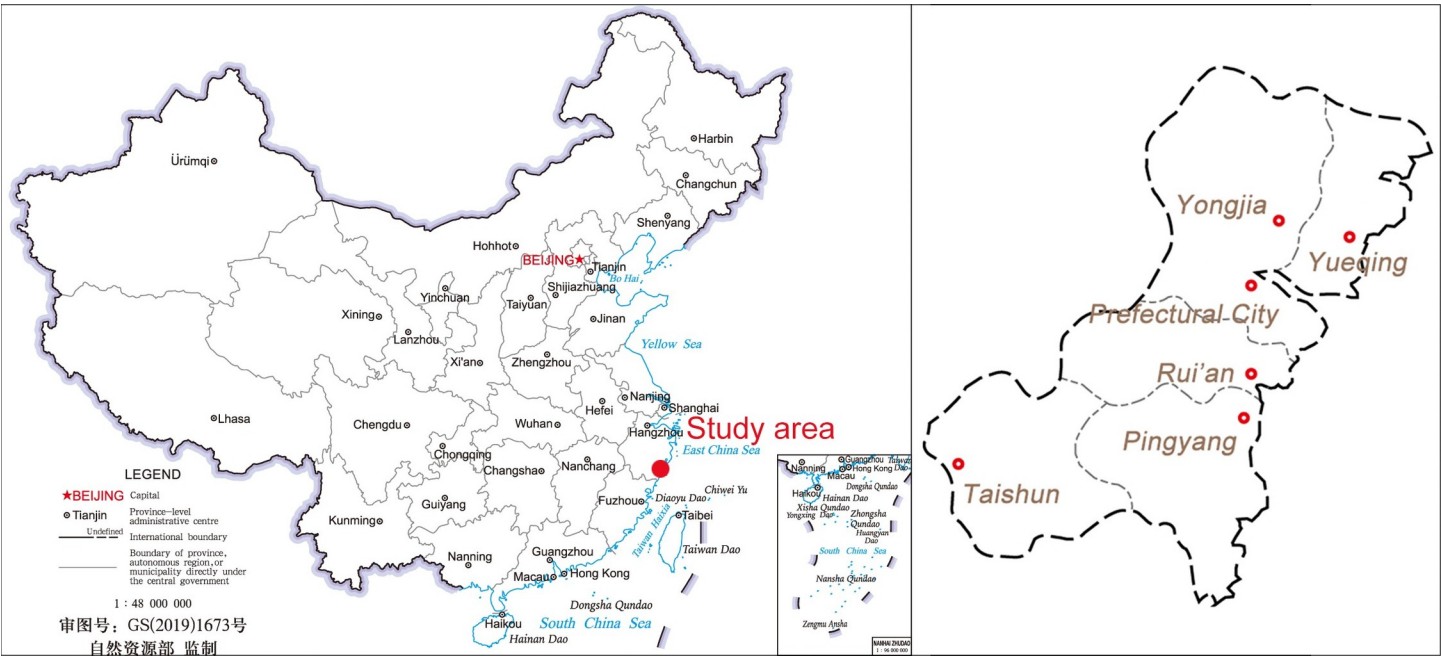

**Fig 1. Location and administrative divisions of the study area.** The map of China (left) was provided by Ministry of Natural Resources of China (Approval number: GS (2019)1673).

Hence, comprehensive local chronicles and coastal military monographs of Wenzhou also lays a foundation for this quantitative study.

This study divided Wenzhou into 5 regions by county, and made a comparative study on the information transmission efficiency in different areas of Wenzhou in the Ming Dynasty, so as to facilitate the understanding of its deployment characteristics and underlying reasons.

### Study object

The information transmission system of Wenzhou in Ming Dynasty could be divided into two sub-systems, beacon system and post system, which included three facilities units: beacon tower (烽燧, *Fengsui*), post station (驿, *Yi*) and urgent delivery station (急递铺, *Pu*). Beacon and post systems were not only independent but also collaborative, taking charge of transmitting information and connecting the front with rear in the coastal wars (Fig 2). They are the indispensable parts of the coastal military defense in Ming Dynasty.

The main functions of the beacon towers were to observe the enemy situation and deliver the warnings and alarms, so sound and light were heavily relied to transmit information. They echoed with each other at a distance. When noticing that the enemy was approaching, the guard would immediately set off the fire to report it to the neighboring beacon towers on the left and right, and the military intelligence would be passed on one by one until the end of the transmission. In this way, the enemy's situation could be quickly transmitted to the central military departments, contributing to quickly mobilizing the garrisons to fight against the enemy forces. However, this method might fail in cloudy and foggy days, so guns were also shot in the Ming Dynasty in addition to fire and smoke, especially in the case of poor visibility and hindered observation. According to the number of invading enemies and urgency degrees, transmission was conducted by different ways. In 1466, the law mandated that: if the number of enemies was less than 100, set off one fire and one gun; if the number of enemies was about

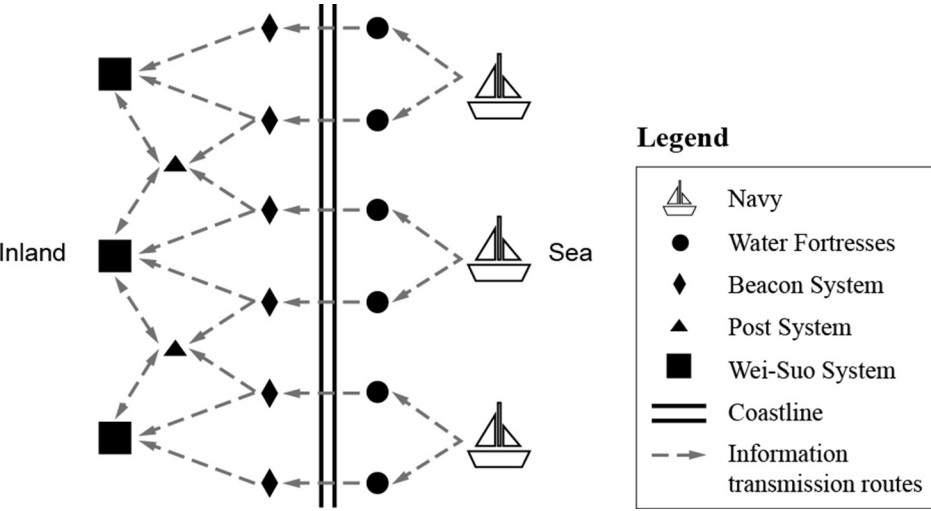

**Fig 2. Coastal defense hierarchical diagram.**

500, set off two fires and two guns; if over 1000, set off three fires and three guns; if over 5000, set off four fires and four guns; if over 10000, set off five fires and five guns [19]. In this way, the fires and guns were increased repeatedly to make the military information delivered more quickly and accurately.

Post stations and urgent delivery stations mainly aimed to transmitting official documents, and which mode to choose was mainly decided by the importance of the documents. The Ming Government divided documents into "common affairs" and "important affairs". Ordinary official documents were classified as "common affairs" and were transmitted by urgent delivery stations; in contrast, "important affairs" mainly included war reports, emergency military affairs, famine reports, disaster reports, and military supplies requests, which were generally transmitted by post stations [20].

In Wenzhou, where the waterway was developed, post route was connected by land and water. Post stations were equipped with boats to deliver, or sedan chairs for escorting ambassadors. There must be an official seal when using the post station and it must be checked by the local government. Seals were divided into two types, namely kanhe (勘合) and huopai (火牌) [3]. Kanhe was widely used, which could be held by the Ministry of War and the government. While huopai was generally used by the Ministry of War, and could only be dispatched by military use. Once the official documents arrived at the post station, they will be relayed immediately.

Urgent delivery stations relied on the couriers to deliver common documents by walking. When official documents arrived at the station, no matter how many pieces they were, day and night, in freezing winter and sultry summer, rain and shine, the couriers must deliver them immediately without detention. *Da Ming Hui Dian* (大明会典) recorded that "the courier held a splint, a bell, a tassel gun, a stick and a notebook", and "ring to deliver" [21]. When it came to the next station, the arrival time must be written down on the notebook, and there will be corresponding punishment if the courier was late, which was extremely strict.

In addition, Wei-Suo system (卫所, Wei city and Suo city, which were the military defense settlements in the Ming Dynasty) were key nodes and major receiving units in the process of information transmission, which would respond to the information delivered. All military information was ultimately handed over to the officers and soldiers in Wei city and Suo city. After receiving the information, they could quickly set up troops to fight. Therefore, Wei-Suo system help to guarantee the efficiency of the transmission process and serve as one evaluation indicator of the study.

## Data sources

All the data of the information transmission system used in this study are taken from *Chou Hai Tu Bian* (筹海图编) [22] and the local chronicles of Wenzhou in the Ming and Qing Dynasties. *Chou Hai Tu Bian* was a coastal military atlas compiled by Ruozhen Zheng (郑若曾) during Jiajing (嘉靖) period of Ming Dynasty. The atlas detailed the number and names of the Wei cities, Suo cities and beacon towers in the coastal areas, and presented the maps of the coastal topography, which could be rated the most comprehensive historical record for the coastal military facilities in the Ming Dynasty. Local chronicles recorded the geography, history, characters, customs, politics, economy, culture, education, transportation, military and other aspects of each region, including the names, routes and probable locations of post stations and urgent delivery stations, providing lots of useful information for the descendents to know about the post system of Wenzhou. Since this study focuses on the Ming Dynasty, the local chronicles of Wenzhou in this period are mainly adopted to extract data, but for some regions whose local chronicles of Ming Dynasty were lost or incomplete, those in the Qing Dynasty are used as a supplement (Table 1).

According to the data extracted, there were about 11 Wei cities and Suo cities, 53 beacon towers, 5 post stations, 117 urgent delivery stations, 1 post route and 12 urgent delivery routes in Wenzhou during the Ming Dynasty (Tables 2 and 3).

## Data visualization

For the convenience of calculation, the above historical data needs to be localized and visualized. The spatial data of each unit were inputted into 30m resolution Digital Elevation Model (DEM) map [23] through ArcGIS software, based on the administrative maps of Wenzhou in the Ming Dynasty depicted in *The Historical Atlas of China* (中国历史地图集) [24] (Fig 3).

**Table 1. Local chronicles statistics.**

| Regions | Dynasties | Local Chronicles |
|---|---|---|
| Wenzhou | Ming Dynasty | Cong Zhang. Wenzhou Prefecture Annals in Jiajing Period. Vol. 1: the Post System. (张璁.嘉靖温州府志.卷之一:驿传.) |
| | Ming Dynasty | Rizhao Tang, Guangyun Wang. Wenzhou Prefecture Annals in Wanli Period. Vol. 3: the Post System. (汤日昭,王光蕴.万历温州府志.卷之三:驿传.) |
| | Ming Dynasty | Zan Wang, Fang Cai. Wenzhou Prefecture Annals in Hongli Period. Vol. 2: the Post System. (王瓒,蔡芳.弘治温州府志.卷之二:邮传.) |
| Yongjia | Ming Dynasty | Shugao Wang, Yingchen Wang. Yongjia County Annals in Jiajing Period. Vol. 3: the Delivery Stations. (王叔杲,王应辰.嘉靖永嘉县志.卷之三:铺舍) |
| | Qing Dynasty | Baolin Zhang, Fen Wang. Yongjia County Annals in Guangxu Period. Vol. 3: the Post Stations and Delivery Stations. (张宝琳,王棻.光绪永嘉县志.卷之三:驿站,铺舍.) |
| Yueqing | Ming Dynasty | Unknown. Yueqing County Annals in Yongle Period. Vol. 4: the Delivery Stations. (作者不详.永乐乐清县志.卷之四:铺舍.) |
| | Qing Dynasty | Dengyun Li, Kun Chen. Yueqing County Annals in Guangxu Period. Vol. 3: the Post System. (李登云,陈珅.光绪乐清县志.卷之三:邮传.) |
| Rui'an | Ming Dynasty | Ji Liu, Chuo Zhu. Rui'an County Annals in Jiajing Period. Vol. 2: the Delivery Stations. (刘畿,朱绰.嘉靖瑞安县志.卷之二:铺舍.) |
| | Qing Dynasty | Yu Zhang. Rui'an County Annals in Qianlong Period. Vol. 2: the Delivery Stations. (章昱.乾隆瑞安县志.卷之二:铺舍.) |
| Pingyang | Qing Dynasty | Qian Sun, Shuxiu Xu, Nanying Zhang. Pingyang County Annals in Qianlong Period. Vol. 4: the Post System. (孙谦,徐恕修,张南英.乾隆平阳县志.卷之四:邮传.) |
| Taishun | Qing Dynasty | Guoyuan Zhu, Tingqi Zhu. Taishun County Annals in Yongzheng Period. Vol. 3: the Delivery Stations. (朱国源,朱廷琦.雍正泰顺县志.卷之三:铺舍.) |

**Table 2. Wei cities, Suo cities, beacon towers, post stations and urgent delivery stations statistics.**

| Regions | Items | Details |
|---|---|---|
| Yongjia | Wei Cities & Suo Cities | Wenzhou Wei, Ningcun Suo |
| | Beacon Towers | Huangshipu, Shagou, Shacun, Qijia, Changsha, Jiujia |
| | Post Stations (Yi) | Xiangpu Yi |
| | Equipment | Xiangpu Yi was equipped with an administrator, a clerk, 6 postal staff, 5 boats, 30 boatmen, 12 sedan chairs and 28 sedan bearers. |
| | Urgent Delivery Stations (Pu) | Head Pu of Wenzhou, Guanghua Pu, Xiaxian Pu, Lintou Pu, Shangwu Pu, Jiangnan Pu, Xialong Pu, Sangxi Pu, Xiaodan Pu, Jiaoyang Pu, Chengnan Pu, Cihu Pu, Quyu Pu, Puzhou Pu, Guacai Pu, Maozhu Pu, Shangwan Pu, Front Pu of Ningcun Suo, Nanmen Pu, Changsha Pu, Little Guishan Pu |
| | Equipment | 77 couriers |
| Yueqing | Wei Cities & Suo Cities | Panshi Wei, Panshi Suo, Puqi Suo |
| | Beacon Towers | Zhang'ao, Shajiao, Sanyu, Chi'ao, Yangtian, Shuangfeng, Rituan, Qitou, Pingshan, Yushan, Baisha, Dongmen, Nanpu, Haitang, Huaqiao, Lou'ao, Xiashantou, Qiantang, Shuangdoumen |
| | Post Stations (Yi) | Guantou Yi, Lingdian Yi, Xi'ao Yi, Yao'aoling Yi |
| | Equipment | Guantou Yi was equipped with an administrator, a clerk, 5 postal staff, 5 boats, 30 boatmen. Lingdian Yi was equipped with an administrator, a clerk, 6 postal staff, 12 sedan chairs and 28 sedan bearers. Xi'ao Yi was equipped with an administrator, a clerk, 5 postal staff, 5 boats, 30 boatmen, 12 sedan chairs and 28 sedan bearers. Yao'aoling Yi was equipped with an administrator, a clerk, 4 postal staff, 12 sedan chairs and 28 sedan bearers. |
| | Urgent Delivery Stations (Pu) | Head Pu of Yueqing, Guantou Pu, Tianxian Pu, Huhuang Pu, Tangxia Pu, Shichuan Pu, Front Pu of Panshi Wei, Baisha Pu, Dalin Pu, Xinshi Pu, Wushi Pu, Dajing Pu, Lanyu Pu, Chenggang Pu, Yitou Pu, Huangshan Pu, Panshan Pu, Shizhen Pu, Front Pu of Puqi Suo, Huwu Pu, Changshan Pu, Sanjiang Pu, Pingfeng Pu, Cai'ao Pu, Tiaotou Pu, Little Chenggang Pu, Little Huangshan Pu |
| | Equipment | 81 couriers |
| Rui'an | Wei Cities & Suo Cities | Rui'an Suo, Hai'an Suo, Shayuan Suo |
| | Beacon Towers | Lengshui, Songfu, Xiankou, Fenghuo |
| | Urgent Delivery Stations (Pu) | Head Pu of Rui'an, Feiyun Pu, Ma'ao Pu, Xianju Pu, Sizhuang Pu, Zi'ao Pu, Litang Pu, Dongshan Pu, Dingtian Pu, Guishan Pu, Shayuan Pu, Qianqiao Pu, Xiang'ao Pu, Wuchi Pu, Tuanyu Pu, Shipai Pu, Panshan Pu, Gexi Pu, Wangyu Pu, Guanyan Pu, Dayang Pu, Tankun Pu, Huanglou Pu, Guixi Pu, Taihu Pu, Huangshan Pu |
| | Equipment | 79 couriers |
| Pingyang | Wei Cities & Suo Cities | Jinxiang Wei, Pingyang Suo, Puzhuang Suo |
| | Beacon Towers | Bantang, Jianshan, Baiqi, Maji, Fenghuang, Maotou, Shangyang, Biwan, Donggang, Lingmen, Dongshan, Mengwan, Lantou, Pacao, Dianshan, Fuquan, Banling, Fengrui, Xuanzhong, Siqiu, Nanbao, Lei'ao, Jianshan, Shijia |
| | Urgent Delivery Stations (Pu) | Head Pu of Pingyang, Ying'en Pu, Wanquan Pu, Changshan Pu, Caidian Pu, Xiangkou Pu, Baoqiao Pu, Fenghuo Pu, Lusi Pu, Xiakou Pu, Guanyuan Pu, Houwei Pu, Qixi Pu, Shuangpai Pu, Pumen Pu, Zhuangshi Pu, Dayi Pu, Yulin Pu, Xiankou Pu, Mocheng Pu, Xiaodu Pu, Tangxia Pu, Hengdu Pu, Lingxi Pu, Xichen Pu, Sizhou Pu, Jiangkou Pu, Huangxiang Pu, Shitang Pu, Fenshui Pu |
| | Equipment | 83 couriers |
| Taishun | Urgent Delivery Stations (Pu) | Head Pu of Taishun, Chikeng Pu, Shangren Pu, Chendai Pu, Hongkou Pu, Zhoukeng Pu, Xiage Pu, Fangcun Pu, Lin'ao Pu, Baishuiji Pu, Fang'ao Pu |
| | Equipment | 33 couriers |

**Table 3. Post routes and urgent delivery routes statistics.**

| Items | Regions | Origins | Destinations | Routes |
|---|---|---|---|---|
| Post Routes | Yongjia, Yueqing | Prefectural City | Boundary of Taizhou Fu | Xiangpu Yi→Xi'ao Yi→Guantou Yi→Yao'aoling Yi→Lingdian Yi |
| Delivery Routes | Yongjia | Prefectural City | Boundary of Rui'an | Head Pu of Wenzhou→Chengnan Pu→Cihu Pu |
| | | Prefectural City | Boundary of Qingtian | Head Pu of Wenzhou→Guanghua Pu→Xiaxian Pu→Lintou Pu→Shangwu Pu→Jiangnan Pu→Xialong Pu→Sangxi Pu→Xiaodan Pu→Jiaoyang Pu |
| | | Prefectural City | Boundary of Yueqing | Head Pu of Wenzhou→Guacai Pu |
| | | Prefectural City | Hai'an Suo | Head Pu of Wenzhou→Quyu Pu→Puzhou Pu→Maozhu Pu→Shangwan Pu→Front Pu of Ningcun Suo→Nanmen Pu→Changsha Pu→Little Guishan Pu |
| | Yueqing | Yueqing Town | Boundary of Huangyan County | Head Pu of Yueqing→Baisha Pu→Dalin Pu→Xinshi Pu→Wushi Pu→Lanyu Pu→Shizhen Pu→Chenggang Pu→Huangshan Pu→Yitou Pu→Dajing Pu→Panshan Pu |
| | | Dalin Pu | Boundary of Taiping County | Dalin Pu→Front Pu of Puqi Suo→Changshan Pu→Sanjiang Pu→Pingfeng Pu→Cai'ao Pu→Tiaotou Pu→Little Chenggang Pu→Little Huangshan Pu→Huwu Pu |
| | | Yueqing Town | Panshi Wei | Head Pu of Yueqing→Tianxian Pu→Huhuang Pu→Tangxia Pu→Shichuan Pu→Guantou Pu→Front Pu of Panshi Wei |
| | Rui'an | Rui'an Town | Boundary of Yongjia | Head Pu of Rui'an→Ma'ao Pu→Xianju Pu→Sizhuang Pu→Zi'ao Pu→Litang Pu |
| | | Rui'an Town | Boundary of Yongjia | Head Pu of Rui'an→Dongshan Pu→Dingtian Pu→Guishan Pu |
| | | Rui'an Town | Boundary of Pingyang | Head Pu of Rui'an→Feiyun Pu→Shayuan Pu |
| | | Feiyun Pu | Boundary of Taishun | Feiyun Pu→Qianqiao Pu→Xiang'ao Pu→Wuchi Pu→Tuanyu Pu→Shipai Pu→Panshan Pu→Gexi Pu→Wangyu Pu→Guanyan Pu→Dayang Pu→Tankun Pu→Huanglou Pu→Guixi Pu→Taihu Pu→Huangshan Pu |
| | Pingyang | Pingyang Town | Boundary of Rui'an | Head Pu of Pingyang→Ying'en Pu→Wanquan Pu |
| | | Pingyang Town | Boundary of Fuding County | Head Pu of Pingyang→Changshan Pu→Dayi Pu→Caidian Pu→Xiaodu Pu→Hengdu Pu→Tangxia Pu→Lingxi Pu→Xichen Pu→Xiangkou Pu→Sizhou Pu→Fenghuo Pu→Fenshui Pu |
| | | Pingyang Town | Puzhuang Suo | Head Pu of Pingyang→Xiankou Pu→Mocheng Pu→Jiangkou Pu→Baoqiao Pu→Lusi Pu→Xiakou Pu→Guanyuan Pu→Yulin Pu→Shitang Pu→Houwei Pu→Shuangpai Pu→Qixi Pu→Huangxiang Pu→Zhuangshi Pu→Pumen Pu |
| | Taishun | Taishun Town | Boundary of Rui'an | Head Pu of Taishun→Chikeng Pu→Shangren Pu→Chendai Pu→Hongkou Pu→Zhoukeng Pu→Xiage Pu→Fangcun Pu→Lin'ao Pu→Baishuiji Pu→Fang'ao Pu |

There are several ways to obtain geographic locations. For the existing units, GPS positioning can be employed for investigation. Some units that have vanished can be accurately localized by using national cultural relics census data, or the locations identified by previous researchers [15, 16]. For those units which are non-existent and untraceable, their approximate geographical scopes can be generalized first based on historical materials; then, their current locations can be identified by searching their names in the modern administrative map, because many ancient names are still used as place names, such as "Guacai Pu" (挂彩铺) in Ming Dynasty and "Guacai Village" (挂彩村) today. If the place name can not be matched, it can be inferred from the locations of known units and the transmission route. After that, all data are proofread to get the probable locations.

Although not all the positioning data are precise, they can still accurately reflect the spatial deployment relationships of the information transmission units in Wenzhou of Ming Dynasty, and provide further analysis materials for this study.

## Evaluation model establishment

Information transmission efficiency refers to the maximum amount of information that can be processed by the transmission network in a unit time [25]. Therefore, it is necessary to

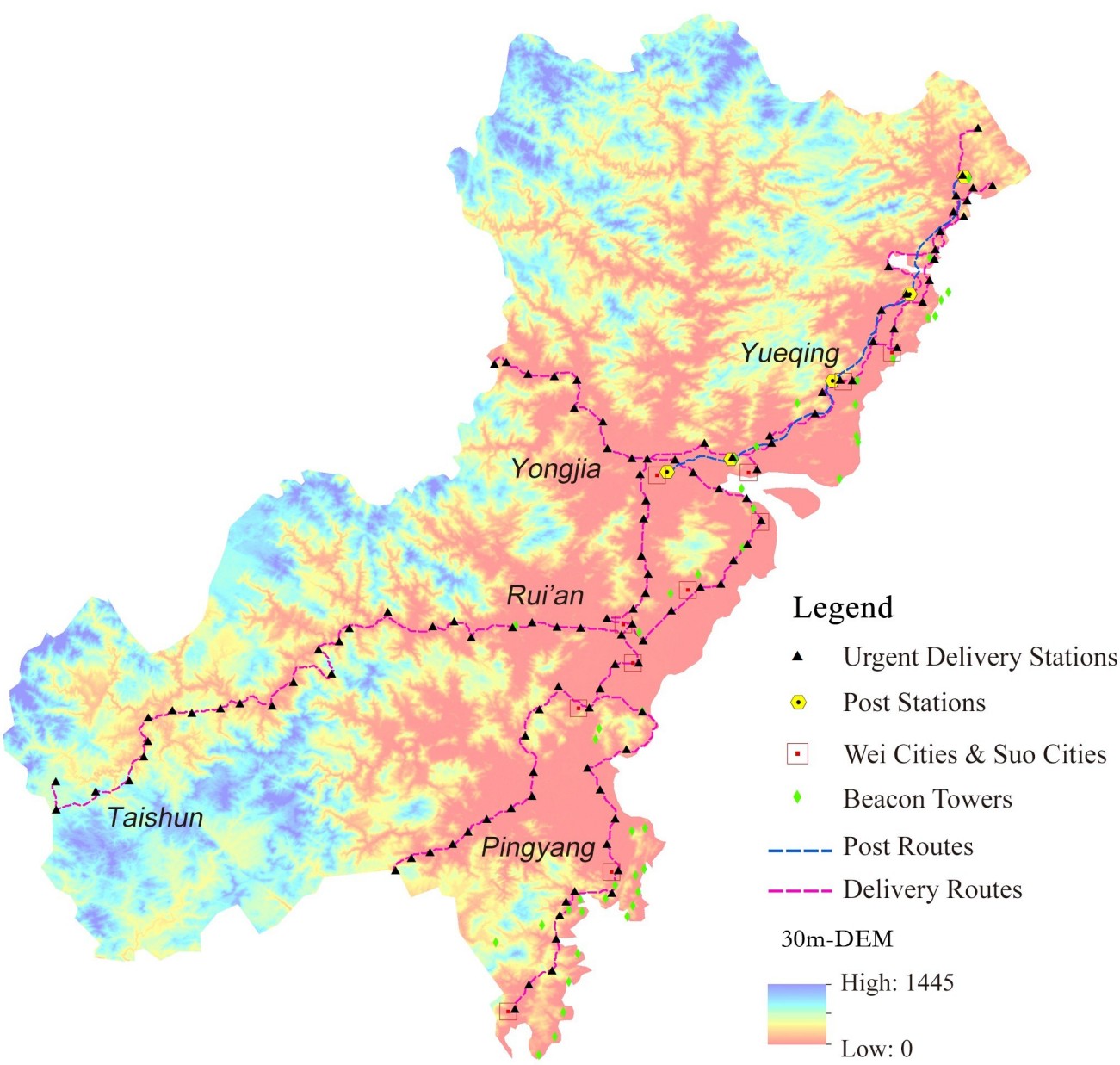

**Fig 3. Spatial distributions of Wei cities, Suo cities, beacon towers, post stations and urgent delivery stations.**

judge whether the process from being transmitted to being received and being responded is efficient and unobstructed. Hence, the number, jurisdiction and distance of each type of transmission units are the key elements in the measurement. This study necessitates the building of a multi-variable hierarchy evaluation model to reasonably evaluate the importance of each indicator.

**Hierarchy evaluation model.** When constructing the hierarchy evaluation model of the information transmission system, the problems should be methodized, and the evaluation influencing factors should be classified and listed hierarchically.

In beacon and post systems of the Ming Dynasty, Wei and Suo cities was important nodes in the process of information transmission. In this study, the influencing factors of information transmission efficiency can be divided into 3 groups, namely, the distribution of Wei-Suo

system, beacon system and post system respectively. Secondly,, further classification is made based on the function and transmission mode of each system.

There are two indicators about Wei-Suo system. One is the number of Wei and Suo cities in this region. The greater the number is, more easily the information will be transmitted. The other one is the average distance of cities in this region. The closer the distance, the shorter the route of transmission and the faster the response.

Since beacon system relied on sound and light to transmit information, the setting of beacon towers must be conducive to the formation of a coherent information flow within the effective reachable range of vision or hearing, so as to transmit enemy situation effectively. Therefore, there are there indicators about beacon system, the number, visual area and average distance of beacon towers respectively.

For post system, it relied tools and couriers to transmit. Therefore, post stations and urgent delivery stations need to be considered separately from the perspectives of quantity, average equipment, average distance and average jurisdiction area. It should be particularly noted that post stations were sparsely distributed in Wenzhou and that only Yueqing was found with more than 1 post station. Thus, it is impossible to measure by the average distance and average jurisdiction area of the post stations, so only the number of post stations is considered in this part. To sum up, there are six indicators about post system, the number and average equipment of post stations, and the number, average equipment, average distance and average jurisdiction area of urgent delivery stations respectively.

According to above-mentioned analysis, a hierarchy evaluation model is established for the information transmission efficiency of Wenzhou in the Ming Dynasty (Fig 4). Taking the information transmission efficiency of Wenzhou in Ming Dynasty as the target (A), the evaluation indicators were divided into two layers. The first layer contains three factors, namely $A = \{a_1, a_2, a_3\}$; the second layer incorporates eight factors, namely $a_1 = \{a_{11}, a_{12}\}$, $a_2 = \{a_{21}, a_{22}, a_{23}\}$, $a_3 = \{a_{31}, a_{32}, a_{33}, a_{34}, a_{35}, a_{36}\}$.

**Judgment matrix.**    However, these evaluation indicators are important to different degrees. In order to accurately reveal their importance, it is necessary to estimate the relative importance of each factor by constructing a judgment matrix and obtaining the weight of each indicator. In this study, the Numbers 1–9 and their reciprocals are used as scales to define the judgment matrix (Table 4).

In order to ensure the accuracy of the results, 10 experts in the study of Ming and Qing history are invited to judge the importance of each indicator in the hierarchy evaluation model, and the average value is adopted as the judgment result to construct the judgment matrix. The judgment matrix $P$ in the first layer is constructed, as shown in Table 5. The judgment matrixs $P1$, $P2$ and $P3$ in the second layer are also constructed, as shown in Tables 6–8 respectively.

Next, the weight value of each judgment matrix needs to be solved. Taking $P \leftarrow \{a_1, a_2, a_3\}$, the first judgment matrix, as an example, the calculation procedures are as follows:

$$P = \begin{bmatrix} 1 & 1/2 & 1/3 \\ 2 & 1 & 1/2 \\ 3 & 2 & 1 \end{bmatrix}$$

① Calculate the product of each row of numbers in the judgment matrix $P$, and get $Mi$.
② Calculate the cubic root of $Mi$, and get $\overline{W}_i$.
③ Normalize $\overline{W} = (\overline{W}_1, \overline{W}_2, \overline{W}_3)$ by using the following equation.

$$W_i = \overline{W}_i / (\sum_{i=1}^{3} \overline{W}_i)$$

Then, $W = (W_1, W_2, W_3)$ is the weight value of judgment matrix $P$.

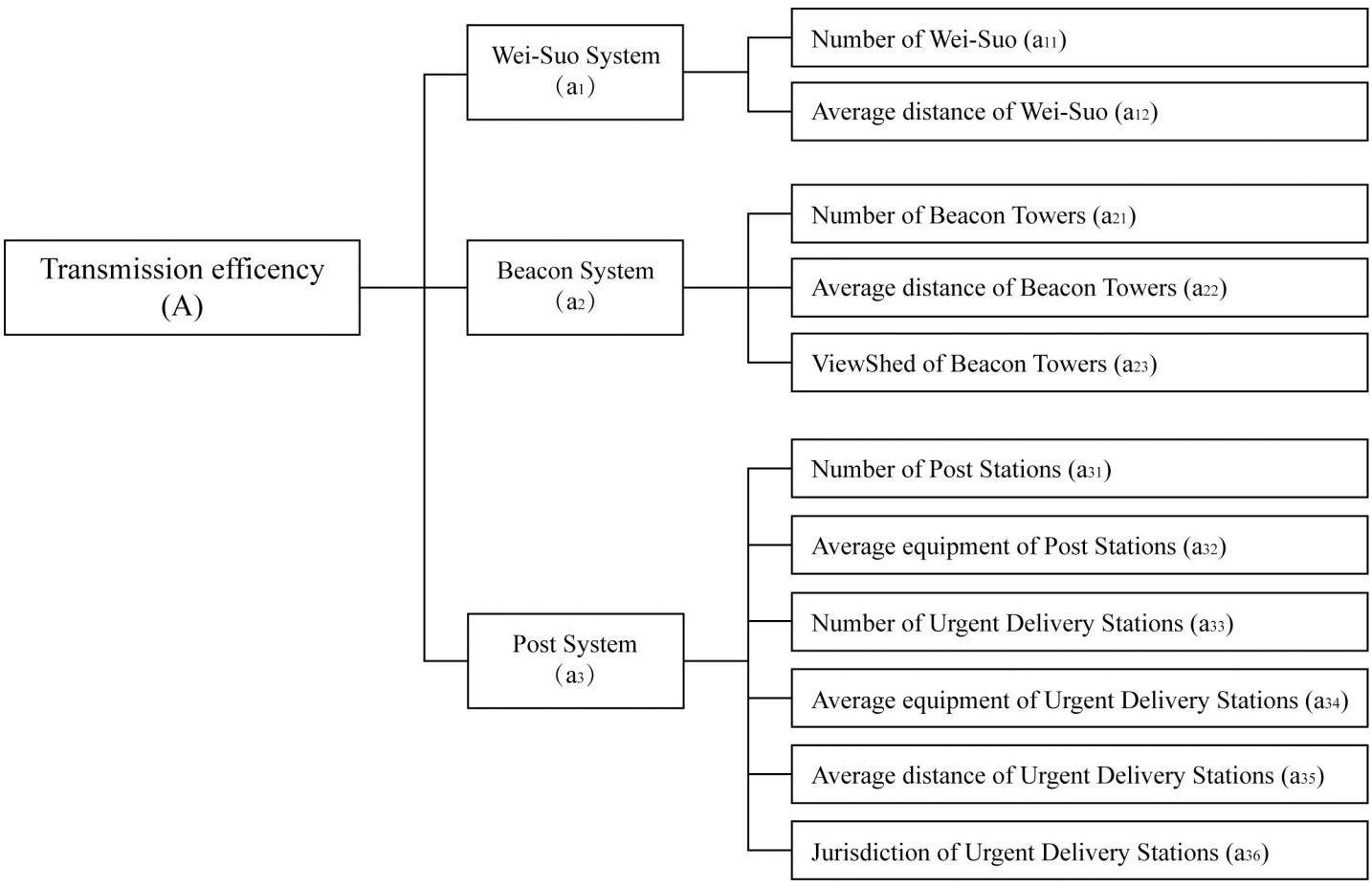

**Fig 4. Hierarchy evaluation model of the information transmission efficiency.**

According to the above steps, the weight value of each judgment matrix can be obtained. The weight value of matrix $P \leftarrow \{a_1, a_2, a_3\}$ is $(W_1, W_2, W_3) = (0.1634, 0.2970, 0.5396)$; the weight value of matrix $P_1 \leftarrow \{a_{11}, a_{12}\}$ is $(W_{11}, W_{12}) = (0.5, 0.5)$; the weight value of matrix $P_2 \leftarrow \{a_{21}, a_{22}, a_{23}\}$ is $(W_{21}, W_{22}, W_{22}) = (0.3333, 0.3333, 0.3333)$; the weight value of matrix $P_3 \leftarrow \{a_{31}, a_{32}, a_{33}, a_{34}, a_{35}, a_{36}\}$ is $(W_{31}, W_{32}, W_{33}, W_{34}, W_{35}, W_{36}) = (0.1, 0.1, 0.2, 0.2, 0.2, 0.2)$.

**Table 4. The fundamental scale of absolute numbers [26].**

| Intensity of importance | Definition |
|---|---|
| 1 | The former and the latter are equally important |
| 3 | The former is slightly more important than the latter |
| 5 | The former is obviously more important than the latter |
| 7 | The former is strongly more important than the latter |
| 9 | The former is extremely more important than the latter |
| 2, 4, 6, 8 | The intermediate value of the above adjacent judgments |
| Reciprocals | If activity $i$ has one of the above nonzero numbers assigned to it when compared with activity $j$, then $j$ has the reciprocal value when compared with $i$ |

**Table 5. $P \leftarrow \{a_1, a_2, a_3\}$.**

| $P$ | $a_1$ | $a_2$ | $a_3$ |
|---|---|---|---|
| $a_1$ | 1 | 1/2 | 1/3 |
| $a_2$ | 2 | 1 | 1/2 |
| $a_3$ | 3 | 2 | 1 |

**Consistency test.** As for whether the weight distribution above is reasonable or not, the consistency test of the judgment matrix is also needed. The test equation is as follows:

$$CR = CI/RI$$

where $CR$ is the random consistency ratio of the matrix, and $CI$ is the general consistency index of the matrix. The calculation equation of $CI$ is given as follows:

$$CI = (\lambda_{max} - n)/(n - 1)$$

where $\lambda_{max}$ is the maximum eigenvalue of the matrix and can be calculated by the following equation:

$$\lambda \text{max} = \sum_{i=1}^{n} \frac{(PW)i}{nWi}$$

$RI$ is the average random index of the judgment matrix which depends on $n$ shown in Table 9.

If $CR$ is less than 0.1, or if $\lambda_{max} = n$ and $CI = 0$, then the matrix can be considered as having an acceptable consistency, and the judgement scales of the matrix are meaningful. Otherwise, the judgment scales in matrix $P$ should be reviewed.

According to the above procedures, calculation results are shown as follows. In matrix P, CR = 0.0079<0.1; in matrix $P_1$, $\lambda_{max}$ = 2, CI = 0; in matrix $P_2$, $\lambda_{max}$ = 3, CI = 0; in matrix $P_3$, $\lambda$max = 6, CI = 0. The calculation results show that the judgment matrices listed have acceptable consistency and the weight values are effective.

## Data quantification

The spatial characteristics of the information transmission system in Wenzhou of the Ming Dynasty need to be quantified, which can be used as the attribute values of the evaluation indicator of the hierarchy evaluation model. Based on the visualized information transmission system of Wenzhou in Ming Dynasty, the spatial characteristics of this system are analyzed by using the spatial analysis tool ArcGIS. Major methods used are as follows: Nearest Neighbor Analysis for the calculation of the average distance between the two units; ViewShed analysis for calculating the visible area of beacon towers (Fig 5); Cost Path analysis for restoring the routes and calculating the average distance between urgent delivery stations (Fig 6); Voronoi Diagram for the calculation of average jurisdiction area of the urgent delivery stations (Fig 7).

**Score quantification.** Due to the inconsistent units of attribute values of these evaluation indicators, in order to unify the fractional magnitude, the attribute value of each evaluation

**Table 6. $P_1 \leftarrow \{a_{11}, a_{12}\}$.**

| $P_1$ | $a_{11}$ | $a_{12}$ |
|---|---|---|
| $a_{11}$ | 1 | 1 |
| $a_{12}$ | 1 | 1 |

**Table 7.** $P_2 \leftarrow \{a_{21}, a_{22}, a_{23}\}$.

| $P_2$ | $a_{21}$ | $a_{22}$ | $a_{23}$ |
|---|---|---|---|
| $a_{21}$ | 1 | 1 | 1 |
| $a_{22}$ | 1 | 1 | 1 |
| $a_{23}$ | 1 | 1 | 1 |

indicator needs to be converted into the hundred-mark score. To be specific, the attribute values of $a_{11}$, $a_{21}$, $a_{23}$, $a_{31}$ and $a_{32}$ are proportional to their scores, while the attribute values of $a_{12}$, $a_{22}$, $a_{33}$ and $a_{34}$ are inversely proportional to their scores.

If the attribute value is proportional to the score, the following equation is adopted:

$$F = \frac{a_n(f_i)}{a_n(f_{\max})} \times 100$$

If the attribute value is inversely proportional to the score, the equation is expressed as follows.

$$F = \frac{a_n(f_{\min})}{a_n(f_i)} \times 100$$

where $F$ is the quantified score after quantization, $a_n(f_i)$ is the attribute value of the $i$-th object in the evaluation indicator $a_n$; $a_n(f_{max})$ and $a_n(f_{min})$ are the maximum and maximum attribute values of the evaluation indicator $a_n$, respectively.

After the quantization, the sore $F$ falls into the range of 0 to 100.

## Results

Through the above calculations and analyses, the evaluation indicator scores for the information transmission efficiency of Wenzhou in the Ming Dynasty are shown in Table 10. Combining the scores with the weight values for calculation, the final evaluation results are shown in Table 11.

The results showed that the information transmission efficiency of different regions in wenzhou in Ming dynasty varied from high to low: Pingyang > Yueqing > Rui'an > Yongjia > Taishun. In details, the indicator values of Pingyang and Yueqing were at a high level, where the transmission efficiency of Pingyang's beacon system was the highest, and that of Yueqing's post system was also the highest. The indicator values of Rui'an and Yongjia were at a similar level, and the transmission efficiency of their post systems was in the middle, while that of the beacon systems was rather low. But for the transmission efficiency of Wei-Suo system, Rui'an was the highest. Taishun had no Wei city, Suo city and beacon tower, so it was the area with the lowest score in all indicators.

**Table 8.** $P_3 \leftarrow \{a_{31}, a_{32}, a_{33}, a_{34}\}$.

| $P_3$ | $a_{31}$ | $a_{32}$ | $a_{33}$ | $a_{34}$ | $a_{35}$ | $a_{36}$ |
|---|---|---|---|---|---|---|
| $a_{31}$ | 1 | 1 | 1/2 | 1/2 | 1/2 | 1/2 |
| $a_{32}$ | 1 | 1 | 1/2 | 1/2 | 1/2 | 1/2 |
| $a_{33}$ | 2 | 2 | 1 | 1 | 1 | 1 |
| $a_{34}$ | 2 | 2 | 1 | 1 | 1 | 1 |
| $a_{35}$ | 2 | 2 | 1 | 1 | 1 | 1 |
| $a_{36}$ | 2 | 2 | 1 | 1 | 1 | 1 |

**Table 9. RI values [27].**

| n | 1 | 2 | 3 | 4 | 5 | 6 | 7 | 8 | 9 |
|---|---|---|---|---|---|---|---|---|---|
| **RI** | 0 | 0 | 0.58 | 0.90 | 1.12 | 1.24 | 1.32 | 1.41 | 1.45 |

## Discussion

By comparing the scores of each region with the distribution map, the information transmission system of Wenzhou in Ming Dynasty is strong in coastal, southern and northern regions and weak in inland and central regions.

From the distribution map of information transmission system (Fig 3), it can be clearly seen that most of the Wei cities, Suo cities, beacon towers, post stations and urgent delivery stations were constructed in coastal areas. On one hand, the coastal terrain of Wenzhou was relatively flat, which was suitable for the construction of various facilities. On the other hand,

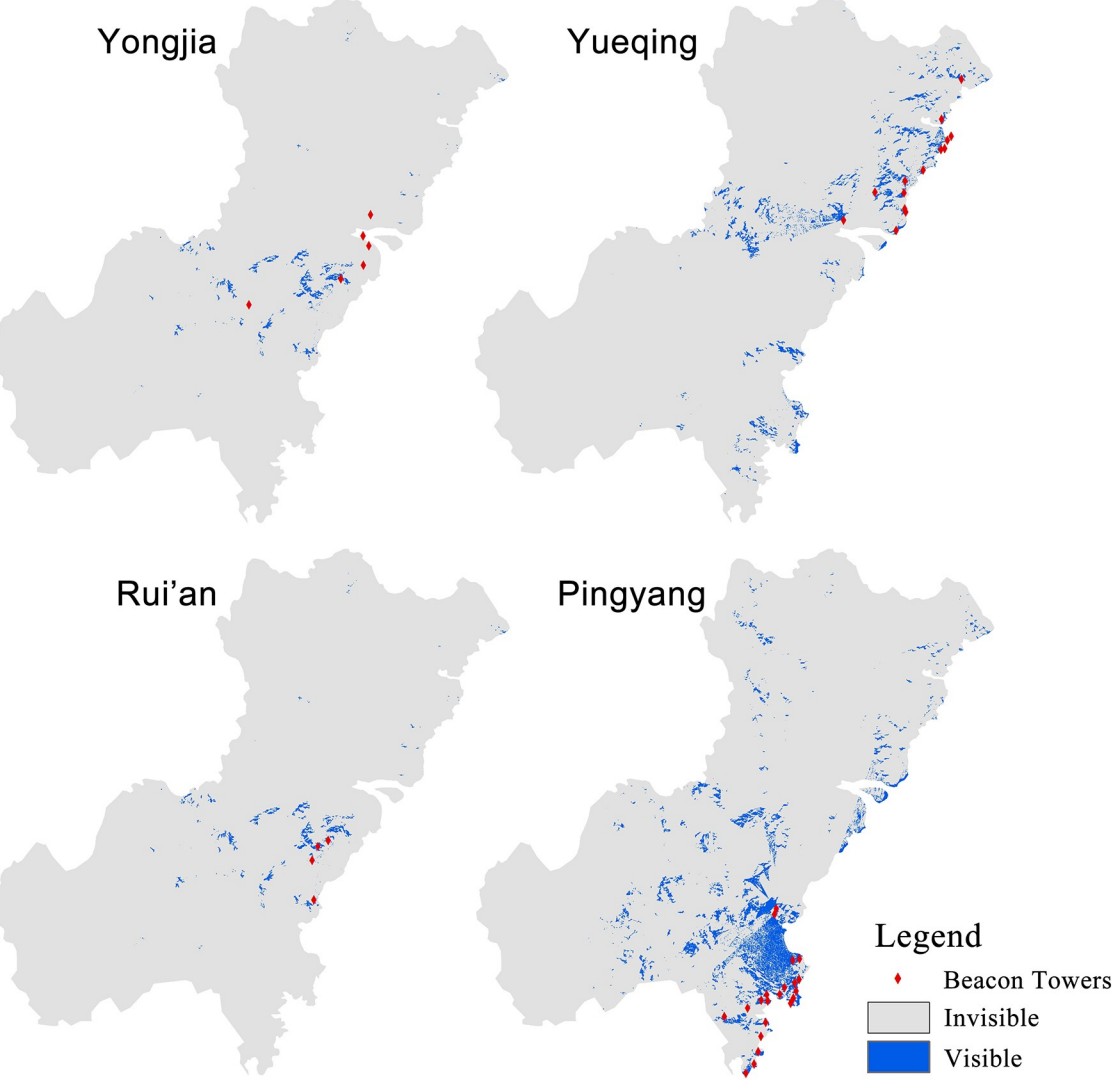

**Fig 5. ViewShed of the beacon towers in different regions.**

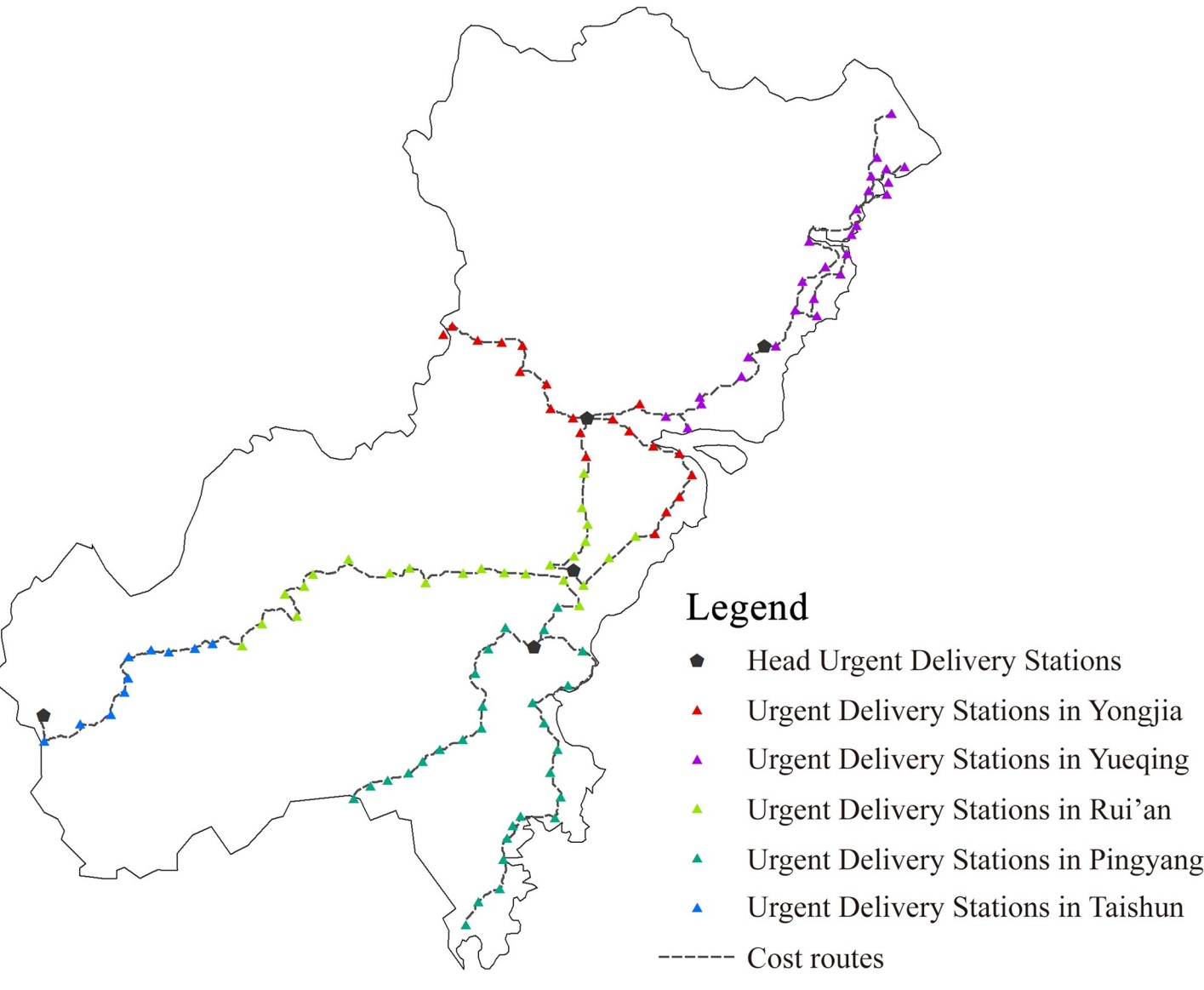

**Fig 6. Cost routes of the urgent delivery stations.**

the coastal defense of Wenzhou focused on the coastal line, which needed the support of a strong transmission system to ensure the unobstructed information flow. However, the inland areas were mostly hilly ones with rugged roads, which were not conducive to long-distance transmission, and there were few wars. Therefore, only a small number of urgent delivery stations were set up to deliver the information when necessary.

Being strong in southern and northern regions and weak in central region, Wenzhou's information transmission system centered on the prefectural city and radiated to the surrounding areas. This had much to do with the centralization of power in Ming Dynasty. In order to realize autocracy, the transmission network of Ming Dynasty centered in the capital and radiated around. For the whole nation, capital was the political and economic center, the transportation hub and the aggregation of the transmission system. The transmission system started from the national capital and radiated to the provincial capitals in different directions,

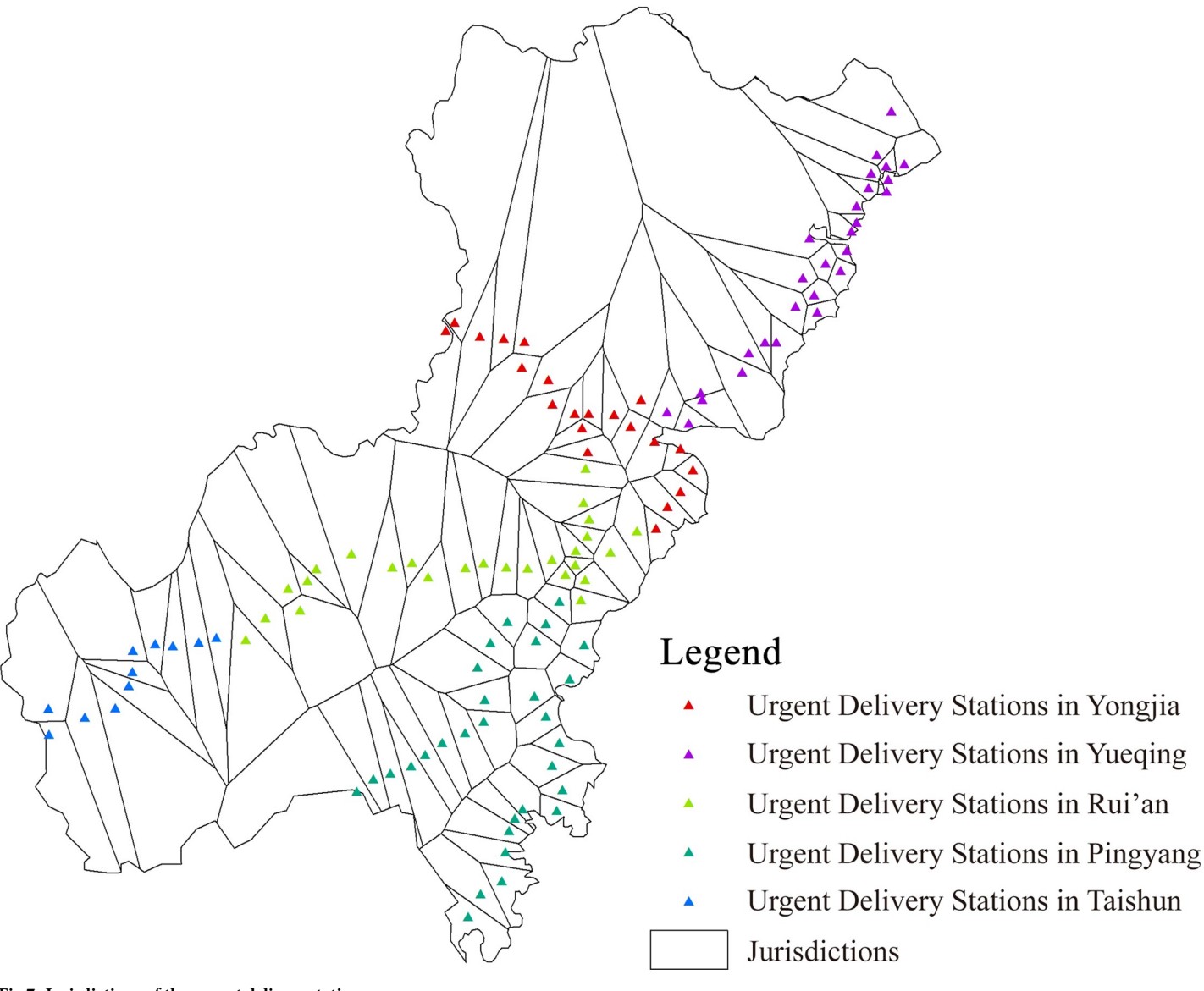

**Fig 7. Jurisdictions of the urgent delivery stations.**

forming a government-level transmission network; then it radiated to the prefectures and connected with the post routes of neighboring provinces, forming a province-level transmission network; and so on. The farther away from the prefectural city, the stronger the transmission efficiency. Only in this way could the managers in the prefectural city quickly control the situation of the whole territory and eliminate the estrangement and block between the central government and the remote areas to the greatest extent.

The transmission efficiencies in the southern and northern regions were also different. Both Pingyang's beacon system and Yueqing's post system were found with a strong transmission efficiency. Why is it the case? Because the reason lies in that Pingyang was directly adjacent to Fujian Province, which was the main route for Japanese pirates to enter Fujian from Zhejiang Province. It not only related to the safety of Zhejiang Province, but also affected the overall situation of Fujian Province, and even the situation of the whole southeastern coastal

**Table 10. Evaluation index scores statistics.**

| | Items | Yongjia | Yueqing | Rui'an | Pingyang | Taishun |
|---|---|---|---|---|---|---|
| $a_{11}$ | Attribute Value | 2 | 3 | 3 | 3 | 0 |
| | Score | 67 | 100 | 100 | 100 | 0 |
| $a_{12}$ | Attribute Value (m) | 17754.17 | 12732.64 | 7956.81 | 26463.34 | 0 |
| | Score | 45 | 62 | 100 | 34 | 0 |
| $a_{21}$ | Attribute Value | 6 | 19 | 4 | 24 | 0 |
| | Score | 25 | 79 | 17 | 100 | 0 |
| $a_{22}$ | Attribute Value (m) | 9689.24 | 5001.7 | 9328.73 | 2950.08 | 0 |
| | Score | 30 | 59 | 32 | 100 | 0 |
| $a_{23}$ | Attribute Value (km$^2$) | 141.23 | 264.79 | 80.8 | 679.82 | 0 |
| | Score | 21 | 54 | 12 | 100 | 0 |
| $a_{31}$ | Attribute Value | 1 | 4 | 0 | 0 | 0 |
| | Score | 25 | 100 | 0 | 0 | 0 |
| $a_{32}$ | Attribute Value | 17 | 11.5 | 0 | 0 | 0 |
| | Score | 100 | 68 | 0 | 0 | 0 |
| $a_{33}$ | Attribute Value | 21 | 27 | 26 | 30 | 11 |
| | Score | 70 | 90 | 87 | 100 | 37 |
| $a_{34}$ | Attribute Value | 3.67 | 3 | 3.04 | 2.77 | 3 |
| | Score | 100 | 82 | 83 | 75 | 82 |
| $a_{35}$ | Attribute Value (m) | 3588.09 | 2954.83 | 3710.70 | 3850.20 | 3613.88 |
| | Score | 82 | 100 | 80 | 77 | 82 |
| $a_{36}$ | Attribute Value (km$^2$) | 163.58 | 106.13 | 123.53 | 82.43 | 195.09 |
| | Score | 50 | 78 | 67 | 100 | 42 |

area. Moreover, it was a place where Japanese pirates frequently invaded. A large number of beacon towers could ensure that the information could be quickly transmitted to the garrisons after discovering the enemy. However, the coastline of Yueqing in the north was blocked by Yuhuan Island, which did not directly rush out to the ocean. On one hand, the sight of the beacon towers must be affected. On the other hand, the invasion frequency of Japanese pirates there was lower than that of Pingyang. Therefore, the transmission efficiency of beacon system in Pingyang was stronger than that of Yueqing. As for the post system, the northern area of Yueqing was adjacent to Taizhou, and the terrain was relatively flat with a waterway. However, there was a strait between the southern area of Pingyang and Fujian Province. Therefore, Yueqing was the best way for Wenzhou to connect with other prefectures. The only post road in Wenzhou was located in Yueqing, which undertook the task of delivering important documents.

Generally speaking, the characteristics of the information transmission efficiency of Wenzhou in Ming Dynasty basically corresponded to the historical environment, which fully reflected that the planning of the information transmission system was considered cautiously

**Table 11. Final scores statistics.**

| Items | Yongjia | Yueqing | Rui'an | Pingyang | Taishun |
|---|---|---|---|---|---|
| Wei-Suo System | 56 | 81 | 100 | 67 | 0 |
| Beacon System | 25.33 | 64 | 20.33 | 100 | 0 |
| Post System | 72.9 | 86.8 | 63.4 | 70.4 | 48.6 |
| Total | 51.41 | 77.27 | 61.24 | 79.13 | 16.2 |

in this period, finally contributing to the formation of an excellent and efficient system. The efficiency variance also suggests that the distribution of information transmission system is not only affected by geographical conditions, but also takes into account the actual military requirements. Thus it can be seen that the Wenzhou's information transmission system played an important role in the military defense of Ming Dynasty. Wenzhou was the key defense area of Zhejiang Province and Fujian Province, so that the developed information transmission system was extremely important for coastal defense. Only officers and soldiers received the frontline intelligence in time can they issue the correct order of fighting. Thus the government attached great importance to the transmission system of coastal areas. Apart from delivering military information, the information transmission system also made a crucial contribution in the transmission of military supplies.

In addition, the information transmission system is also regarded as centralization strengthening in feudal countries. The central government used all resources to ensure the smooth movement of taxes and personnel, and the high-speed circulation of all kinds of information by managing the transmission system. Moreover, with the help of the transmission system, the local government could been informed and implemented the imperial decree in time, which greatly eliminated the estrangement and obstruction between the central government and local regions. Thus, the rule of the central government could extend to every corner of the country so as to ensure the long-term stability.

To conclude, the information transmission system played an important role in the military and politics of ancient China. Being well-organized, completed, convenient, safe and reliable, it undertook the task of information dissemination in politics, economy, culture, military and other fields, which was of great significance in the communication between the central governments and local regions.

## Conclusions

By using quantitative comparative method, a hierarchy evaluation model for Wenzhou in the Ming Dynasty is established in this study, which explores the differences in the information transmission efficiency of different regions. The model can be used to not only evaluate the overall information transmission efficiency, but also identify the differences among Wei-Suo system, beacon system and post system. Model demonstration reveals that it brings out the results that cannot be obtained by intuitive feeling, and clearly reveals the distribution characteristics of Wenzhou's information transmission system in Ming Dynasty. Besides, it is characterized by clear methods, easy operation, and high repeatability. Therefore, the model is highly suitable for evaluating the information transmission system of Wenzhou in Ming Dynasty, and can also be widely used in the study of ancient information transmission system, so as to facilitate people's understanding of history.

The results for Wenzhou in the Ming Dynasty indicate that the information transmission efficiency of different regions vary from high to low: Pingyang > Yueqing > Rui'an > Yongjia > Taishun. Separately, Rui'an is found with the highest transmission efficiency of Wei-Suo system; Pingyang boasts the highest transmission efficiency of beacon system; Yueqing has the highest transmission efficiency in terms of post system. The results are basically in agreement with the historical environment, suggesting that the spatial distribution of the information transmission system is not only affected by geographical conditions, but also takes into account the actual military requirements. Therefore, it can be proved that the information transmission system in the coastal areas of Ming Dynasty is designed to serve for military defense to a large extent.

## Supporting information

**S1 File. Scanned version of *Chou Hai Tu Bian*.**
(TIF)

**S2 File. Jurisdiction area of urgent delivery stations.**
(XLS)

**S3 File. Scanned version of local chronicles.**
(ZIP)

**S4 File. Average nearest neighbor calculation results.**
(ZIP)

**S5 File. ViewShed area of beacon towers.**
(ZIP)

## Author Contributions

**Conceptualization:** Yukun Zhang, Bei Wu, Lifeng Tan, Jiayi Liu.

**Data curation:** Bei Wu, Lifeng Tan.

**Formal analysis:** Bei Wu.

**Funding acquisition:** Lifeng Tan.

**Investigation:** Bei Wu, Lifeng Tan.

**Methodology:** Bei Wu, Lifeng Tan.

**Project administration:** Lifeng Tan.

**Resources:** Yukun Zhang.

**Software:** Bei Wu, Jiayi Liu.

**Supervision:** Bei Wu, Jiayi Liu.

**Validation:** Bei Wu, Jiayi Liu.

**Visualization:** Bei Wu, Jiayi Liu.

**Writing – original draft:** Bei Wu.

**Writing – review & editing:** Yukun Zhang, Lifeng Tan.

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
