## [Decision Letter · Decision Letter 0]

11 Jan 2021

PONE-D-20-35409

Quantitative research on the efficiency of ancient information transmission system：A case study of Wenzhou in the Ming Dynasty

PLOS ONE

Dear Dr. Wu,

Thank you for submitting your manuscript to PLOS ONE. After careful consideration, we feel that it has merit but does not fully meet PLOS ONE’s publication criteria as it currently stands. Therefore, we invite you to submit a revised version of the manuscript that addresses the points raised during the review process.

We look forward to receiving your revised manuscript.

Kind regards,

Chi-Hua Chen, Ph.D.

Academic Editor

PLOS ONE

Journal Requirements:

2.We note that Figure(s) S1, 1, 3, 5, 6 and 7 in your submission contain map images which may be copyrighted. All PLOS content is published under the Creative Commons Attribution License (CC BY 4.0), which means that the manuscript, images, and Supporting Information files will be freely available online, and any third party is permitted to access, download, copy, distribute, and use these materials in any way, even commercially, with proper attribution. For these reasons, we cannot publish previously copyrighted maps or satellite images created using proprietary data, such as Google software (Google Maps, Street View, and Earth). For more information, see our copyright guidelines: http://journals.plos.org/plosone/s/licenses-and-copyright.

a.    You may seek permission from the original copyright holder of Figure(s) S1, 1, 3, 5, 6 and 7 to publish the content specifically under the CC BY 4.0 license. 

Reviewers' comments:

Reviewer's Responses to Questions

**Comments to the Author**

1. Is the manuscript technically sound, and do the data support the conclusions?

Reviewer #1: Partly

2. Has the statistical analysis been performed appropriately and rigorously? 

Reviewer #1: I Don't Know

3. Have the authors made all data underlying the findings in their manuscript fully available?

Reviewer #1: Yes

4. Is the manuscript presented in an intelligible fashion and written in standard English?

Reviewer #1: Yes

5. Review Comments to the Author

Reviewer #1: This paper is very interesting because of its objective: to measure the efficiency of ancient Chinese information transmission systems (civil and military) by taking the case of the postal relay system at the time of the Chinese Ming Dynasty (1368-1644) in the Wenzhou region, a region divided into five counties. Although the authors allude to Chinese antiquity and advance the idea that, compared to previous systems, the information transmission systems under the Ming were '' very developed '' (line 45), this claim is not proven, especially compared to those of Yuan, Song and Tang Dynasties. Moreover, when we observe the postal system of the various Chinese dynasties, we are struck by its enormous global development and more generally the development of techniques in ancient China (see Needham, Joseph, Science and civilization in China, 1954-2015, Cambridge University Press). We are thus surprised to read line 40 that “In ancient China, where technology was underdeveloped...”! We do not agree, on the contrary, technologies in China were very developed for those times! This is what the research and work of J. Needham and his collaborators has demonstrated. The maps and tables are extremely interesting because they represent and highlight the physical organization of the Chinese postal system in this region, and that is nothing compared to the extent of the Chinese empire under the Ming dynasty.

By the number of post relays, signal towers and relays for the distribution of urgent information, these maps show, extremely well, the extreme political and military importance given by the Chinese authorities to the organization of the transmission of information. The work and technical data of this article are therefore invaluable.

But a number of critical remarks should be made. First of all, it is regrettable that, despite a rich bibliography, no mention is made of works relating to the history of the Chinese postal systems which shows a continuity of this organisation from antiquity to pre-modern times (see Gazagnadou, Didier, 2016, The diffusion of a postal relay system in Premodern Eurasia, Kimé ed., Paris, chap. I and II and the bibliography). The Ming postal system inherits from that of the Yuan, Song, Tang, etc. As for the efficiency of the system in the transmission of information, the authors should have mentioned the speed of circulation of postal mails to link the towns together (Gazagnadou, chap. I) and that, for the land post, the efficiency of the transmission depends on the type of horses and the number of kilometres separating the different relays (Minetti, Alberto, 2003, Efficiency of the Equine of the postal system, Nature, n ° 246, December) and of course on the regional topography. The authors could have provided a map and idea of the area covered by postal routes equipped with the different types of relay and post stations. This point is extremely important and should be reworked by the authors.

To conclude, this article with very interesting materials and perspectives should be enriched by more technical information, in particular on the '' beacon tower (烽燧, Fēng suì) '', the '' post station (驿, Yì) '' and the '' urgent delivery station (急 递 铺, Jí dì pù) '' and this should be possible in particular by making full use of all the information contained in the local chronicles of Wenzhou from the Ming period that the authors quote in their tables.

Finally, the authors do not insist enough the links between the transmission of the military information and the politics and the pivotal role of this system of transmission of information and the crucial role of horse in this organisation for the efficiency of the system.

6. PLOS authors have the option to publish the peer review history of their article (what does this mean?). If published, this will include your full peer review and any attached files.

Reviewer #1: No

---

## [Author Response · Author response to Decision Letter 0]

7 Mar 2021

Dear editor and reviewer,

On behalf of all the contributing authors, I would like to express our sincere appreciations of your letter and reviewers’ constructive comments concerning our article entitled “Quantitative research on the efficiency of ancient information transmission system：A case study of Wenzhou in the Ming Dynasty” (Manuscript No.:PONE-D-20-35409). These comments are all valuable and helpful for improving our article. According to the academic editor and reviewer’s comments, we have made extensive modifications to our manuscript and supplemented extra data to make our results convincing. In the revised version, changes to our manuscript were all highlighted within the document by using “Track Changes”. For the places that not been modified according to the comments, we also made explanations. Point-by-point responses to the academic editor and reviewer are listed below this letter.

Response to academic editor’s comments:

1)Please ensure that your manuscript meets PLOS ONE's style requirements, including those for file naming.

Response: We modified the manuscript according to the formatting sample of PLOS ONE to ensure that all formats and file naming have been adjusted to conform to the requirements indicated in PLOS ONE's submission guidelines.

2)We note that Figure(s) S1, 1, 3, 5, 6 and 7 in your submission contain map images which may be copyrighted....We require you to either (1) present written permission from the copyright holder to publish these figures specifically under the CC BY 4.0 license, or (2) remove the figures from your submission.

Response: We are very sorry for our negligence of the copyright issue in the process of using the pictures. Some of the pictures mentioned above are self drawn and there is no copyright licensing problem. Others are removed or replaced according to the suggestions.

Figure S1 is not directly related to the manuscript, we just take it as a reference material. In order to avoid unnecessary misunderstanding and copyright disputes, we removed Figure S1 and add the atlas with Figure S1 (Qixiang Tan. The Historical Atlas of China. Sino Maps Press; 1982.) into the reference.

Figure 1 contains two maps, one is the location of Wenzhou in the Ming Dynasty, the other is the administrative map of Wenzhou. Unfortunately, we didn't get in touch with the author of the Ming Dynasty map, so we have to replace this map. The new map of China is provided by Ministry of Natural Resources of China (http://bzdt.ch.mnr.gov.cn). According to the website, the public can browse and download the standard map for free, and the map approval number should be marked when using the standard map directly. We use this map as required. While the administrative map of Wenzhou remains, because it is self drawn with reference to historical materials.

Figure(s) 3, 5, 6 and 7 in the manuscript are self drawn by using the ArcGIS software. All the information contained in these maps are collected from the historical materials, and analyzed by the spatial tools of ArcGIS software.

Response to reviewer’s comments:

1)Although the authors allude to Chinese antiquity and advance the idea that, compared to previous systems, the information transmission systems under the Ming were '' very developed '' (line 45), this claim is not proven, especially compared to those of Yuan, Song and Tang Dynasties.

Response: The transmission system of Ming Dynasty inherited from that of the Yuan Dynasty. Compared with the transmission system of the Yuan, Song and Tang Dynasties, the laws on the transmission system in the Ming Dynasty were more completed, which had made specific provisions on the organization, network distribution, mileage time limit and funding quota. In addition, the post roads in the border areas were hewed out, which made the coverage of the transmission system larger. (see Guangsheng Liu, Meizhuang Zhao. The history of ancient post in China. Posts and Telecommunications Press; 1999.) Therefore, the transmission system of the Ming Dynasty was "very developed". We are very sorry that our expression is not rigorous enough, so we make a supplementary explanation about the development of the transmission system in the Introduction part.

2)We are thus surprised to read line 40 that “In ancient China, where technology was underdeveloped...”! We do not agree, on the contrary, technologies in China were very developed for those times! This is what the research and work of J. Needham and his collaborators has demonstrated.

Response: Thank the reviewer for pointing out this wrong expression. Our original intention of “underdeveloped” refers to the comparison with the convenient and diversified communication tools in modern society. In order to emphasize that the dissemination of information in ancient China could only rely on transmission system built by the government. But it does cause ambiguity, so we delete the modification of “where technology was underdeveloped”.

3)It is regrettable that, despite a rich bibliography, no mention is made of works relating to the history of the Chinese postal systems which shows a continuity of this organisation from antiquity to pre-modern times (see Gazagnadou, Didier, 2016, The diffusion of a postal relay system in Premodern Eurasia, Kimé ed., Paris, chap. I and II and the bibliography).

Response: We sincerely appreciate the valuable comments. As suggested by the reviewer, we have read more documents on the history of Chinese postal system and made a supplement of that in the Introduction part to support this study.

4)As for the efficiency of the system in the transmission of information, the authors should have mentioned the speed of circulation of postal mails to link the towns together (Gazagnadou, chap. I) and that, for the land post, the efficiency of the transmission depends on the type of horses and the number of kilometres separating the different relays (Minetti, Alberto, 2003, Efficiency of the Equine of the postal system, Nature, n ° 246, December) and of course on the regional topography. The authors could have provided a map and idea of the area covered by postal routes equipped with the different types of relay and post stations.

Response: The land information transmission in Ming Dynasty could be divided into two ways, one was the urgent delivery stations relying on couriers to deliver common documents by walking, and the other was the post stations equipped with horses, boats or sedan chairs. In Wenzhou, where the waterway was deveoped, post route was connected by land and water. Post stations were equipped with boats to deliver, or sedan chairs for escorting ambassadors. (see the local chronicles of Wenzhou) We appreciate Minetti and Alberto's research on post horses. They contributed valuable information to the research of postal system. But their findings do not suit for this study, because there was no post horse in Wenzhou. As can be seen from the post equipment, Wenzhou’s land transmission was mainly relied on the urgent delivery stations. If we understand correctly, the reviewer's suggestion is to take into account the impact of post equipment on the study of transmission efficiency. We totally agree with that. Post equipment is indeed a factor which we ignored. Therefore, we have supplemented the data on the equipment of post stations and urgent delivery stations, that is, the number of couriers and boats, and rebuilt the evaluation model, to make the results more rigorous and convincing. In addition, according to the reviewer's suggestion, we also reconstructed the historical map of Wenzhou’s transmission system by adding the postal routes.

5)This article with very interesting materials and perspectives should be enriched by more technical information, in particular on the '' beacon tower (烽燧, Fēng suì) '', the '' post station (驿, Yì) '' and the '' urgent delivery station (急递铺, Jí dì pù) '' and this should be possible in particular by making full use of all the information contained in the local chronicles of Wenzhou from the Ming period that the authors quote in their tables.

Response: Thank the reviewer for the affirmation of this study. In the part of Study object, we supplement the technical information about beacon towers, post stations and urgent delivery stations, especially their transmission mode and organization mode, to support the following discussion.

6)Finally, the authors do not insist enough the links between the transmission of the military information and the politics and the pivotal role of this system of transmission of information and the crucial role of horse in this organisation for the efficiency of the system.

Response: In the Discussion part, we reanalyzed Wenzhou’s information transmission system and discussed the relationship between the information transmission and military and politics, as well as the pivotal role of the transmission of information system. As previously stated, Wenzhou’s post station did not use horse as transmission tool, therefore, we do not insist the role of horse. 

I hope that the changes and explanations we’ve made resolve all your concerns about the article. I’m more than happy to make any further changes that will improve the paper and facilitate successful publication. Thank you very much for your help.

Sincerely,

Bei Wu

---

## [Decision Letter · Decision Letter 1]

12 Apr 2021

Quantitative research on the efficiency of ancient information transmission system：A case study of Wenzhou in the Ming Dynasty

PONE-D-20-35409R1

Dear Dr. Wu,

We’re pleased to inform you that your manuscript has been judged scientifically suitable for publication and will be formally accepted for publication once it meets all outstanding technical requirements.

Kind regards,

Chi-Hua Chen, Ph.D.

Academic Editor

PLOS ONE

Additional Editor Comments (optional):

Reviewers' comments:

Reviewer's Responses to Questions

**Comments to the Author**

1. If the authors have adequately addressed your comments raised in a previous round of review and you feel that this manuscript is now acceptable for publication, you may indicate that here to bypass the “Comments to the Author” section, enter your conflict of interest statement in the “Confidential to Editor” section, and submit your "Accept" recommendation.

Reviewer #1: All comments have been addressed

2. Is the manuscript technically sound, and do the data support the conclusions?

Reviewer #1: Yes

3. Has the statistical analysis been performed appropriately and rigorously? 

Reviewer #1: Yes

4. Have the authors made all data underlying the findings in their manuscript fully available?

Reviewer #1: Yes

5. Is the manuscript presented in an intelligible fashion and written in standard English?

Reviewer #1: Yes

6. Review Comments to the Author

Reviewer #1: I am very happy with the answers of the authors. The six points that were questioned are carefully and positively addressed by the authors.

7. PLOS authors have the option to publish the peer review history of their article (what does this mean?). If published, this will include your full peer review and any attached files.

Reviewer #1: No

---

## [Editor Report · Acceptance letter]

14 Apr 2021

PONE-D-20-35409R1 

Quantitative research on the efficiency of ancient information transmission system：A case study of Wenzhou in the Ming Dynasty 

Dear Dr. Wu:

I'm pleased to inform you that your manuscript has been deemed suitable for publication in PLOS ONE. Congratulations! Your manuscript is now with our production department. 

Kind regards, 

on behalf of

Professor Chi-Hua Chen 

Academic Editor

PLOS ONE